# Lithography Alignment Techniques Based on Moiré Fringe

**Wenbo Jiang** [1,2,*] **, Huaran Wang** [1,2]**, Wenda Xie** [1,2] **and Zhefei Qu** [1,2]

[1] School of Electrical Engineering and Electronic Information, Xihua University, Chengdu 610039, China
[2] Sichuan Provincial Key Laboratory of Signal and Information Processing, Xihua University, Chengdu 610039, China
***** Correspondence: jiangwenbo@mail.xhu.edu.cn

**Abstract:** In Moiré fringe lithography alignment technology, alignment is realized by monitoring the grating interference fringe image in real-time. The technique exhibits excellent sensitivity to displacement changes and is not easily affected by the gap changes between the mask and silicon wafer. Therefore, this technique is widely used in conventional proximity and contact lithography and new-generation micro- and nanolithography systems. The rapid development of semiconductor and integrated circuit industries, as well as the increasing requirements for the resolution of various nanodevices and systems, have posed new challenges in Moiré fringe lithography alignment technology, which are mainly reflected in alignment accuracy, alignment range, and scheme complexity. In this study, the development history, alignment principle, and overall process of Moiré fringe lithography alignment technology are reviewed; the main factors affecting alignment accuracy are analyzed, and corresponding optimization schemes are provided; and finally, the development trend and research focus of Moiré fringe lithography alignment technology are predicted from the marking structure, alignment scheme, and algorithm processing.

**Keywords:** lithography alignment technology; moiré fringe; alignment accuracy and range; Influencing factors and optimization scheme

## 1. Introduction

The development of advanced integrated circuits and semiconductor chips has followed Moore's Law, which was proposed in 1965. This model relies on the continuous development of micro-nano device manufacturing technology, in which lithography technology plays a crucial role. Ultra large-scale integration process nodes with sizes of 5 nm and below have been developed. Furthermore, conventional lithography techniques are greatly bound by the diffraction limit. Therefore, plasma interference lithography [1], nanoimprint technology [2], extreme ultraviolet lithography [3], and other new generations of lithography technology have been developed.

Lithography alignment technology is one of the three core technologies of the lithography system that directly affect the quality and performance level of lithography products. Existing mainstream lithography alignment methods can be classified into three categories, namely geometric image alignment, light intensity information alignment, and phase information alignment. Among them, the alignment method based on geometric images involves realizing alignment by using the images of alignment marks [4], which has the advantages of a simple system and large alignment range, but its alignment accuracy is low. The alignment method based on light intensity information involves realizing alignment by detecting and capturing the specific diffraction order light intensity of the grating or the maximum critical intensity of the diffraction light [5]. This method exhibits a high signal-to-noise ratio, high accuracy, and excellent real-time alignment. However, the method has poor process adaptability and high requirements for achieving a stable state of the mask and wafer gap. The alignment method based on phase information involves using the phase information of the beat frequency signal generated between the wafer and the mask

alignment marks to achieve alignment [6], and the gap between the alignment marks does not affect the period and phase distribution of Moiré fringes. High alignment accuracy can be achieved by combining this result with fringe processing technology and high-precision interference technology. The alignment method can effectively avoid signal fluctuations or magnetic susceptibility changes caused by the wafer process layer [7]. Furthermore, superior structural complexity and system robustness were achieved. Therefore, as a typical representative of phase information alignment, alignment technologies based on Moiré fringes are widely used in the new generation of lithography systems.

## 2. Generation Mode of Moiré Fringe and Its Development in Lithography Alignment

### 2.1. Generation and Classification of Moiré Fringe

In 1872, Rayleigh revealed the scientific principle behind and the engineering value of Moiré fringes [8], which can be easily and simply generated, have low requirements for light sources, can be used to easily acquire formed images, and have a sensitive response to displacement and deformation. Since then, Moiré fringes have been widely used in medicine, industrial measurement, image encryption, three-dimensional (3D) imaging, lithography, and other fields [9–11].

In essence, Moiré fringes are a modulation phenomenon of the periodic or directional difference of the diffraction grating structure on the incident light intensity. Their light field is affected by the grating type and the superposition mode of the modulated light intensity.

According to the nature of the grating, a Moiré fringe can be categorized into conventional and digital fringes. Conventional Moiré fringe generation depends on the physical grating. According to the optical arrangement of the system, Moiré fringes can be categorized into shadow and projection types [12]. Among them, the shadow-type fringe is produced by the interference between the grating and its shadow, which is widely used in the measurement field. The projection-type fringe is produced by the superposition of two gratings with periodic or directional differences, and it is widely used in lithography alignment. In the digital type of fringe, the digital grating or a captured grating image is used to replace conventional physical grating, or digital image processing technology is used to modulate and combine the grating to generate Moiré fringes.

The superposition modes of light intensity for forming a Moiré fringe can be categorized as addition, subtraction, and multiplication types [13]. Addition and subtraction Moiré fringe images are generated by double-field exposure and interference cancellation and are mostly used in the field of strain measurement. Their structure is complex, and the generation process is more cumbersome than the multiplication type, which is generated by the natural superposition of double gratings, and it contains more independent difference frequency terms than the addition and subtracted types; thus, it can be used to better analyze the grating displacement. Therefore, this type of fringe is mostly used for lithography alignment.

### 2.2. Application Status of Moiré Fringes in Lithography Alignment

Moiré fringes were first applied to lithographic alignment in 1972. In the subsequent 50 years, Moiré fringes developed from physical grating to digital grating; the fringes formed by these two types of gratings are introduced and discussed separately in this section.

2.2.1. Development of Physical Grating

The development process of early physical gratings involves single grating and composite gratings.

(1)　Circular/linear single grating:

In 1972, American scholar M.C. King applied Moiré fringe technology to the proximity lithography alignment system and observed the zero-order Moiré fringes through a microscope for alignment. The result revealed that alignment accuracy was limited by the accuracy of the repetitive stepping motor. Ultimately, 200-nm alignment accuracy was achieved [14]. In 1998, Japanese scholar Shigeru Kawai applied a group of Fresnel zone

plates with various periods to the lithography alignment system, as shown in Figure 1a. The alignment was realized by observing the first-order diffraction light imaging, and alignment accuracy higher than 40 nm was achieved [15]. However, Moiré fringes generated through circular grating exhibit the disadvantages of spectral leakage and the easy occurrence of sampling errors due to their low matching degree with charge-coupled device (CCD) rectangular pixels. Using this method, the extraction and analysis of phase information are difficult, and its alignment accuracy is low. Therefore, this method is mostly used for coarse alignment.

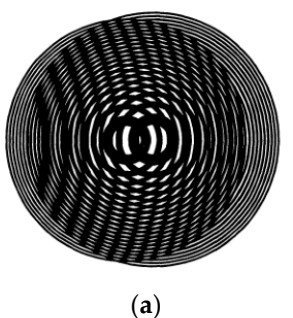
(**a**)

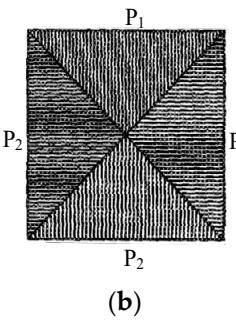
(**b**)

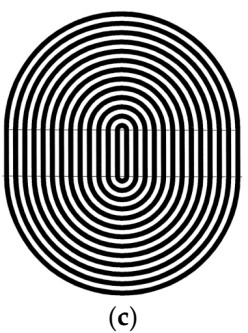
(**c**)

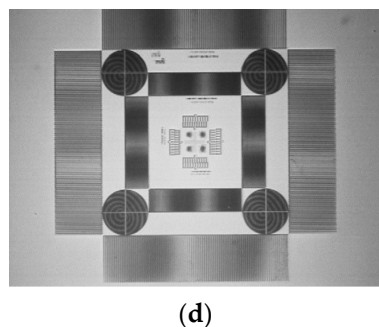
(**d**)

**Figure 1.** Generation method of aligned Moiré fringes in lithography based on physical grating. (**a**) Circular grating. (**b**) Line grating. (**c**) Elongated circular gratings. (**d**) Composite grating.

To solve the technical problems encountered by circular grating Moiré fringe alignment technology, American scholar Flanders first set the period of two groups of gratings to 10-μm linear grating in the lithography alignment system and achieved an alignment accuracy of 100 nm [16]. The results revealed that when the grating period decreased to 1.2 μm, the alignment accuracy was improved to 20 nm. Although the phase change caused by the displacement of the line grating is easy to obtain and analyze, the Moiré fringe generated by linear grating arranged in the same direction can only achieve the detection and calibration of mask–wafer misalignment by comparing the phase between two sets of Moiré fringes averaged along the fringe direction. This direction can be defined as the "vertical direction", which is perpendicular to the longitudinal Moiré fringe.

To realize the alignment of the *x*-axis and *y*-axis directions of the two-dimensional plane at the same time, in 1981 and 1993, MIT scholars Lyszczarz and Smith proposed two combinations of linear gratings [17,18]. Lyszczarz used a line grating orthogonal to the *x*–*y* direction to form a "cross" alignment mark, whereas Smith's team used a rectangular alignment mark composed of four parts of gratings to form an interference pattern with the same spatial period, but images moving in the direction opposite the mask's translation. The alignment marks contained two ($P_1$, $P_2$) pairs for X alignment and Y alignment. The Moiré fringes of each part were combined to achieve a 2D plane multi-directional alignment, mask mark as shown in Figure 1b, swapping the periods of $P_1$ and $P_2$ on the mask mark and converting it into wafer marks.

(2) Composite grating:

To solve the technical problem that the spatial displacement direction cannot be determined by simple linear grating in the lithography alignment system, in 1998, Korean scholar Jong sup Song [19] combined line grating and circular grating. The alignment marks were composed of two semicircular gratings and one line grating with the same period, as shown in Figure 1c. The two semicircular gratings and the center line grating were matched with each other, and the mask and wafer alignment marks were complementary and used in the lithography alignment system. The alignment accuracy was one-tenth of the grating period.

In 2006, Princeton University added four squares and cross coarse marks in the center at the upper left and right corners of entire alignment marks. The three groups of line gratings arranged around them were used as fine alignment marks with a magnification

of 20 times. Two groups of Moiré fringes were formed side by side, which increased magnification by two times, and an alignment accuracy of 20 nm was achieved [20]. In 2007, Austrian scholar Mühlberger used the standard cross structure arranged at four corners for rough alignment and added a circular grating around it to determine the displacement direction; Vernier patterns were employed in order to overcome the ambiguity of linear Moiré structures, as shown in Figure 1d. Finally, four groups of line-grating Moiré structures were used to achieve high-precision alignment with an alignment accuracy of 100 nm [21]. In 2013, Chinese scholar Zhu proposed four-quadrant alignment marks. Each alignment mark includes two groups of line gratings and a cross structure at the center. The cross structure realizes coarse alignment, and the line grating realizes fine alignment, with a measurement error of less than 10 nm [22,23].

### 2.2.2. Development of Digital Grating

High-precision Moiré fringe alignment technology can be achieved by using composite grating alignment marks to realize coarse and fine alignment. However, the complexity of mask manufacturing and the slow speed of traditional mask manufacturing methods, such as e-beam writing to produce physical masks, greatly increases their cost, and the smoothness of the etched curve cannot easily satisfy the requirements of an integrated optical system. The spatial resolution and alignment accuracy of Moiré fringes are limited. Therefore, digital mask lithography technology was proposed to replace conventional physical mask grating with digital grating and generate Moiré fringes using a computer, which simplifies the alignment system and provides the potential for rapid Moiré fringe analysis.

Digital grating generation methods include vibration surface average sampling, external video signal, electronic scanning sampling, the Talbot self-imaging effect, and computer generation and digital micromirror device (DMD) projection, among others. In 1991, the University of Hong Kong proposed logical digital grating to measure the displacement between two gratings. Different logic operations were applied to the two gratings generated by the computer to analyze the phase change caused by their displacement and obtain the displacement distance [24]. In 1998, the National University of La Plata proposed using the Talbot effect to generate grating-like fringes to measure the rotation angle of the measuring object. In 2012, the Institute of Optoelectronic Technology of the Chinese Academy of Sciences used a DMD to project a group of digital gratings for photolithography alignment. Digital gratings were projected onto the silicon wafer and superimposed with the existing physical grating marks to generate Moiré fringes. The alignment scheme is shown in Figure 2; because of the low overall resolution of this alignment system and the grating error of DMD projection, the measurement accuracy was only 65 nm [25].

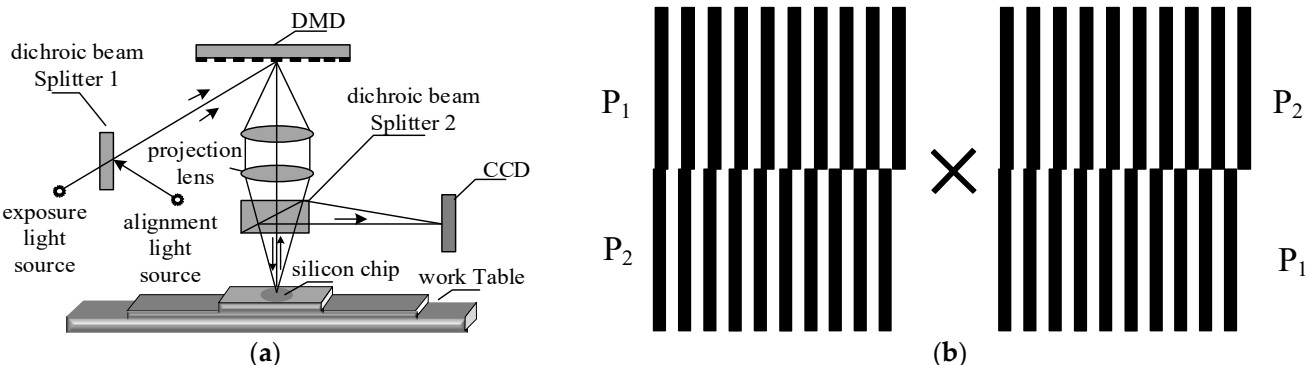

**Figure 2.** Generation method of lithographic aligned Moiré fringe based on digital grating. (**a**) Digital micro-mirror device (DMD) digital grating alignment system. (**b**) Digital grating alignment mark.

*2.3. Summary of Moiré Fringe Lithography Alignment Technology*

The early Moiré fringe lithography alignment technology typically improved alignment accuracy by improving alignment marks. Subsequently, circle and line single grating and the combined grating of the two were performed to obtain both coarse and fine alignment.

With the maturity of grating manufacturing technology and the structure of alignment marks, the focus of research has gradually shifted from the improvement of alignment marks to the optimization of alignment schemes, and novel digital grating and Moiré fringe digital synthesis methods without physical constraints have been developed.

According to much of the literature, the main problems of Moiré fringe lithography alignment technology are its small alignment range, poor Moiré image quality, and inaccurate phase extraction algorithms.

## 3. Moiré Fringe Alignment Principle and Alignment Process

### 3.1. Alignment Principle

According to Section 2.1, the most common mode in Moiré fringes alignment is to generate multiplicative Moiré fringes by superimposing projected line gratings. When the grating is displaced, the displacement of the corresponding Moiré fringe causes the average intensity of a point to change, and it is reflected in the phase distribution of the Moiré fringe image. The actual deviation between the mask and the wafer can be calculated by analyzing the Moiré fringe image. This section introduces the alignment principle of Moiré fringe lithography based on this model.

The actual photolithography alignment process is not always performed in an ideal environment. A modicum of external tilt and in-plane rotation angles exist, which severely affect the Moiré fringe image formed when the subsequent line grating amplifies the lateral displacement. Therefore, the first alignment task is to correct the external tilt and in-plane rotation angles and then calibrate the vertical grid line direction.

#### 3.1.1. Calibration Principle of Out-of-2D Plane Inclination Angle

In the out-of-plane tilt calibration, considering the standard straight line rectangular line grating as an example, the spatial repetition frequencies of the line grating are $f_1$ and $f_2$, and the complex amplitude distribution of the two gratings is as follows:

$$E_1 = \sum_{m=-\infty}^{\infty} A_m \exp[i2\pi f_1 G_1(x,y)] \tag{1}$$

$$E_2 = \sum_{n=-\infty}^{\infty} B_n \exp[i2\pi f_2 G_2(x,y)] \tag{2}$$

In Equations (1) and (2), $A_m$ and $B_n$ represent the Fourier coefficients of the grating, and $G_1(x,y)$ and $G_2(x,y)$ represent the transmission coefficients related to the geometric surface of the grating. The overlapping of two gratings causes multiple-order diffraction on their surfaces, which results in a series of diffraction wave components overlapping each other to form an ordered distribution. Generally, the component passing through the first grating where m-order diffraction occurs and the second grating where n-order diffraction occurs is called (m, n) harmonic. Therefore, the superposition of all harmonic components ultimately determined the field distribution after double grating diffraction, that is,

$$E(x,y) = \sum_{m,n} A_m B_n \exp\{i2\pi[mf_1 G_1(x,y) + nf_2 G_2(x,y)]\} \tag{3}$$

Under normal incidence of the monochromatic plane wave with a wavelength of $\lambda$, diffraction occurs on the surfaces of the two gratings to form an interference field. The lower diffraction order of the interference field is easily imaged. According to Equation (3), the interference field distribution of the two beams (1, 0) and (0, 1) is expressed as follows:

$$E_{(1,0)}(x,y) = A_1 B_0 \exp[i2\pi m f_1 G_1(x,y)] + A_0 B_1 \exp[i2\pi m f_2 G_2(x,y)] \tag{4}$$

Through simplification, the interference intensity distribution can be expressed as follows:

$$I_{(1,0)}(x,y) = \left| E_{(1,0)}(x,y) \right|^2 = I_1 + I_2 + 2\sqrt{I_1 I_2} \cos\left\{ F' \cdot X] + \varphi_0 \right\} \tag{5}$$

In Equation (5), $I_1 = (A_1 B_0)^2$ and $I_2 = (A_0 B_1)^2$ represent the intensities of the two beams, $\varphi_0$ represents the initial phase difference, $X = (x, y)$ represents the coordinates on the XOY plane, and $F'$ represents the spatial frequency vector of the interference field. The relative inclination of the wafer and the mask directly changes the beam deflection direction (corresponding to the frequency vector of the grating) and the phase distribution of the interference field. The frequency vector after deflection is as follows:

$$F' = \frac{1}{\lambda} \left[ \frac{\tan(\theta_1 + 2\delta_\theta)}{\sqrt{1 + \tan^2(\theta_1 + 2\delta_\theta) + \tan^2 2\delta_\omega}} - \sin\theta_2, \frac{\tan 2\delta_\varphi}{\sqrt{1 + \tan^2(\theta_1 + 2\delta_\theta) + \tan^2 2\delta_\omega}} \right] \tag{6}$$

In Equation (6), $\theta_1$ and $\theta_2$ are the diffraction angles of two diffracted beams, $\delta_\theta$ is the deflection angle decomposed into the deflection of the cross-section ZOX, and $\delta_\omega$ is the deflection decomposed into the deflection of the longitudinal section ZOY. The transverse and longitudinal deflection leads to a change in the Moiré fringe frequency vector, which determines the out-of-plane deflection of the mask to the wafer.

### 3.1.2. Principle of In-Plane Rotation Angle Calibration

The frequency and direction of the Moiré fringes are changed when the two gratings are rotated at an angle. This principle can be used to calibrate the in-plane angular displacement of the mask and wafer planes. The periodic structure participating in the overlap is a standard cosine grating with cosine light intensity changes. For the commonly used $(+1, -1)$ stacked grating fringe, the expression of the transmitted light intensity of the two line gratings is as follows:

$$I_1 = 0.5 + 0.5 \cos[2\pi f_1 G_1(x,y)] \tag{7}$$

$$I_2 = 0.5 + 0.5 \cos[2\pi f_2 G_2(x,y)] \tag{8}$$

When the two gratings are overlapped and irradiated with plane waves of unit intensity, the light field distribution after overlapping is as follows:

$$\begin{aligned} I = I_1 \times I_2 = \tfrac{1}{4}\{ &1 + \cos[2\pi f_1 G_1(x,y)] + \cos[2\pi f_2 G_2(x,y)] \\ &+ \tfrac{1}{2}\cos 2\pi [f_1 G_1(x,y) + f_2 G_2(x,y)] \\ &+ \tfrac{1}{2}\cos 2\pi [f_1 G_1(x,y) - f_2 G_2(x,y)] \} \end{aligned} \tag{9}$$

In Equation (9), the last difference frequency term exhibits a lower spatial frequency, which is the fringe term. The fringe distribution is obtained by rotating two standard line gratings. Let the transformation rule of its geometric distribution function be as follows:

$$G_1(x,y) = x\cos a_1 + y\sin a_1 \tag{10}$$

$$G_2(x,y) = x\cos a_2 + y\sin a_2 \tag{11}$$

where $a_1$ and $a_2$ represent the rotation angles of the grating, and $F_v = (\cos a_1 / P_1) - (\cos a_2 / P_2)$ and $F_u = (\sin a_1 / P_1) - (\sin a_2 / P_2)$ represent the frequency of the stacked grating fringe in the $x$ and $y$ directions, respectively. The inclination angle of the Moiré oblique fringe with respect to the horizontal direction is expressed as follows:

$$\omega_{\text{moire}} = \arctan(F_v / F_u) \tag{12}$$

If $a_2 = 0$, that is, if the rectangular coordinates are established based on the grating $G_2$ (without rotation), then $a_1 = \Delta\delta$, where $\Delta\delta$ represents the angular displacement between the two sets of gratings, and the angle between the Moiré fringes formed at this time and the grating $G_1$ is given by

$$\omega_1 = \arctan\left(\frac{P_1 \sin\Delta\delta}{P_1 \cos\Delta\delta - P_2}\right) \tag{13}$$

Similarly, the angle between the Moiré fringe and the grating $G_2$ can be deduced, and the period of the Moiré fringe formed is given by

$$P_{moire} = P_1 P_2 / \sqrt{P_1^2 + P_2^2 - 2P_1 P_2 \cos\Delta\delta} \tag{14}$$

From Equations (13) and (14), the relationship between the pitch of the Moiré fringes and the included angle in the horizontal direction and the angular displacement of the two gratings can be obtained to realize the calibration of the inner rotation angle of the Moiré fringes.

### 3.1.3. Principle of Fine Calibration in the Vertical Grid Line Direction

After the internal and external angular displacements of the two planes were calibrated, the displacement in the face perpendicular to the grating direction was finely calibrated based on the amplification effect of Moiré fringes on small displacement. The line grating shown in Figure 2b is used to align the upper and lower groups of Moiré fringes formed by the mask and the silicon wafer.

When the grating is in the initial coordinates, for the commonly used $(+1, -1)$ level-stacked grating fringes, the corresponding period of generating Moiré fringes is as follows:

$$P_{moire} = 1/(f_1 - f_2) = P_1 P_2 / (P_2 - P_1) \tag{15}$$

When the theoretical shift is $\Delta x$ in the direction perpendicular to the grating line, the phase shifts generated by the upper and lower groups of Moiré fringes are as follows:

$$\phi_{up} = 2\pi\Delta x / P_1, \phi_{bottom} = -2\pi\Delta x / P_2 \tag{16}$$

The phase shift directions of the upper and lower groups of stripes are opposite, and the phase shift difference is as follows:

$$\Delta\phi_{moire} = \phi_{up} - \phi_{bottom} = 2\pi\Delta x(1/P_1 + 1/P_2) \tag{17}$$

The relative movement of the upper and lower groups of stripes is as follows:

$$\Delta X_{moire} = \Delta X_{up} + \Delta X_{bottom} = \frac{P_1 + P_2}{P_2 - P_1}\Delta x \tag{18}$$

Equation (18) shows the amplification effect of the Moiré fringes generated by the line grating on the small displacement between the mask and wafers. The phase relationship between the displacement of the mask to the wafer and the Moiré fringes can be obtained as follows:

$$\Delta x = [\Delta\phi_{moire}/2\pi] \times (P_a/2) = \Delta\phi_{moire}/[2\pi(f_1 + f_2)] \tag{19}$$

where $P_a$ is the average period of two differential gratings, defined as $P_a = 2P_1 P_2 / (P_1 + P_2)$. The displacement can be obtained by extracting the phase of the Moiré fringes, and the lithographic alignment of the Moiré fringes can be completed.

### 3.2. Alignment Process

According to the alignment principle, the process of applying Moiré fringes in lithography alignment includes wafer leveling and focusing, tilt angle calibration between the

mask and wafer planes, in-plane angular displacement correction after two planes are parallel and high-precision axial displacement calibration without angular displacement.

### 3.2.1. Calibration of Out-of-2D Plane Inclination Angle

Before exposure, the wafer should be accurately positioned within the focal depth range of the objective projection lens, and the upper surface of the silicon wafer should be perpendicular to the axis of the lithographic objective lens (or parallel to the ideal focal plane). The resolution of the lithography system can be improved by shortening the wavelength of the light source and increasing the numerical aperture of the objective lens. However, with the resolution greatly increased, the focal depth sharply decreased and the exposure field gradually increased. Lithography leveling and focusing technology has also been developed, which in their early stages used slits or pinhole marks to obtain multi-point heights to realize the field-by-field focusing of the center point of the exposure field after leveling the entire field. However, this technology is now used to detect the intensity or phase of the interference beam, such as in the optical heterodyne method for realizing field-by-field leveling and focusing [26].

Moiré fringes are used for leveling and focusing without complex optical equipment, with high alignment accuracy and high tolerance to environmental factors. In 2014, Chinese scholar Di proposed a four-channel leveling and focusing lithography scheme based on Moiré fringes [27]. Its schematic is shown in Figure 3.

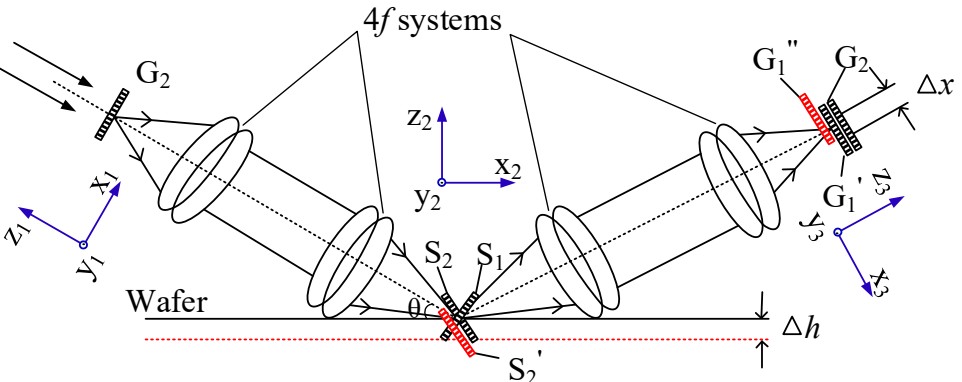

**Figure 3.** Schematic of Moiré fringe leveling and focusing scheme.

Figure 3 only shows the single-channel detection of defocus at a point on the wafer and the grating shown in Figure 2b. The optical system can be simply decomposed into two 4*f* systems, and the focal length of all lenses is *f*. When the diffraction grating beam $G_1$ is reflected on the wafer through the left system and $G_1'$ is imaged on the rear focal plane of the right system, Moiré fringes are generated by the image $G_1'$ and $G_2$. $S_1$ and $S_2$ are the images of $G_1$ before and after reflection on the wafer. The displacement of the wafer at the vertical distance $\Delta h$ will cause $S_2$ to shift to $S_2'$, resulting in the fringe phase change $\Delta\phi_{\mathrm{moiré}}$ and the displacement $\Delta X_{moire}$, which can be calculated as follows:

$$\Delta h = \frac{\Delta x}{2\sin\theta} = \frac{1}{2\sin\theta}\frac{\Delta\phi_{moire}}{2\pi}\frac{P_1 \cdot P_2}{P_1 + P_2} = \frac{1}{2}\frac{\Delta X_{moire}}{2\sin\theta}\left(\frac{|P_1 - P_2|}{P_1} + \frac{|P_1 - P_2|}{P_2}\right) \quad (20)$$

In Equation (20), $\theta$ is the angle between the incident light and the standard wafer, and the phase change caused by the lateral displacement of Moiré fringes can be demodulated to obtain the vertical displacement of the wafer to complete point-by-point focusing. The vertical displacement of the four focal points in the wafer plane can be detected, and the inclination and center defocus of the wafer can be easily calculated by using the heights of the four focal points. This method can achieve the detection accuracy of several nanometers. Through multi-point detection, it can achieve a global inclination error of $10^{-8}$ rad for a wafer with a diameter of 30 cm.

The study revealed that when the deviation between the wafer and the standard plane is more than several microns, the generated Moiré fringes are inclined. Based on this phenomenon, combined with the ability to achieve Moiré fringe calibration of the out-of-plane inclination angle (outlined in Section 3.1.1), the team placed two gratings, as shown in Figure 2b, both on the mask plane. The diffracted light of one grating was reflected by the wafer and formed Moiré fringes with the other grating [28], as shown in Figure 4.

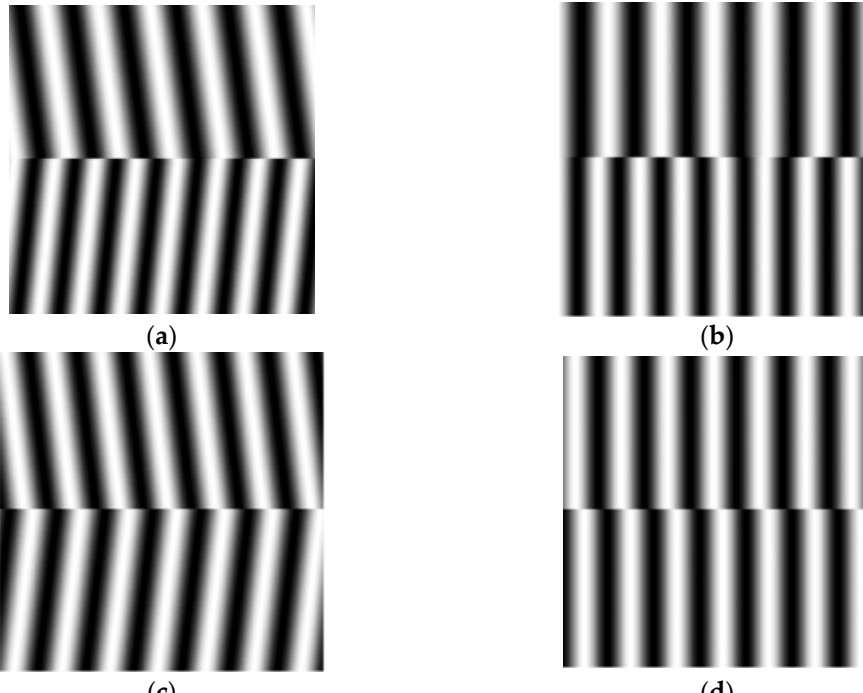

**Figure 4.** Distribution of two sets of fringes when the wafer is tilted. (**a**) Both sections. (**b**) Only the cross-section. (**c**) Only the longitudinal section. (**d**) Leveled.

From Equation (6), the out-of-plane inclination angle decomposed into the transverse cross-section causes the upper and lower parts of the Moiré fringe frequency to change, and the longitudinal offset causes the stripes to be angled.

Although this method associates the mask with the wafer at the out-of-plane tilt, it can only measure single-point plane tilt detection with an accuracy of $10^{-3}$ rad. With this accuracy, there will be hundreds of microns of error at the other end of the wafer. If there are gratings placed at this point, Moiré fringes will not be generated. Therefore, single-point detection of out-of-plane tilt is meaningless for global alignment and may only be used for real-time alignment monitoring; otherwise, multi-point field-by-field detection can achieve the accuracy of today's photolithography.

This method has three problems: the different stripe phenomena caused distinct detection accuracy in the two directions' offset, which affects the overall correction accuracy of the plane. Because of the absorption of reflected light by the surface photoresist and the process layer, the Moiré stripe image is not clear; this can be compensated for to some extent by its excellent image deblurring algorithm [29]. This process should go through coarse positioning, tilt preprocessing, height extraction, and tilt calibration, among other steps, which increases the complexity of the system and introduces system errors.

### 3.2.2. In-Plane Angular Displacement Calibration

After the out-of-plane inclination of the mask to the wafer is calibrated and the two planes are in parallel, the intersecting Moiré fringes are formed because of the angular displacement of the mask to the wafer. The position information obtained by analyzing the

phase difference of Moiré fringes is inaccurate, and the angular displacement between the two planes should be calibrated.

In 2013, the Institute of Optoelectronic Technology, Chinese Academy of Sciences, used circular gratings to detect and correct in-plane angular displacement in the early stage. Two pairs of circular gratings were arranged on the mask and the wafer. First, one group of pre-aligned Moiré fringes was recorded. When the two planes generated in-plane angular displacement, the phase distribution of the diffraction field of the other group of grating changed to generate different Moiré fringe images. The in-plane angular displacement was calculated by the two groups of Moiré fringe images.

The method requires two points on the same plane to be placed with grating, and the phase extraction accuracy of the generated Moiré fringes is not as good as that of line grating. To achieve superior alignment accuracy and simplify the alignment process, the team implemented angular displacement correction using four-quadrant gratings, which were detected only at a single point. In the presence of angular displacement, the resulting Moiré fringes are similar to those shown in Figure 4, with the same spatial frequency of the upper and lower part of the fringes and forming an angle. The relationship between the angle $\Delta\theta_{moire}$ of the Moiré fringes and the angular displacement θ in the grating plane is as follows:

$$\Delta\theta_{moire} = \arctan\left(\cot\delta\theta - \frac{f_1}{f_2}\csc\delta\theta\right) + \arctan\left(\frac{f_2}{f_1}\csc\delta\theta - \cot\delta\theta\right) \tag{21}$$

In the case of using gratings with periods $P_1$ = 4.0 μm and $P_2$ = 4.4 μm, even if the table correction is manually controlled, an angle correction accuracy of $10^{-4}$ rad can be obtained [30].

From Equation (21), the actual angular displacement can be calculated if the included angle $\Delta\theta_{moire}$ of the Moiré fringes is known or if the spatial frequency component of the Moiré fringes on the *x* and *y* axes is detected by phase analysis. However, several problems still occur. Angle detection cannot easily capture the included angle of Moiré fringes and has low accuracy. When detecting the frequency component, the angular displacement is small, and leakage of the frequency spectrum of Moiré fringes occurs on the*x* and *y* axes, which results in a large error. This method can only obtain the angular displacement between the two gratings; the angle between the grating and the worktable (*x*-, *y*-axis) is unknown. Therefore, this method is not suitable for high-precision automatic adjustment worktable to achieve alignment.

A space–frequency decomposition method (SFDM) was then proposed to effectively solve the problems and improve alignment accuracy. In SFDM, the grating shown in Figure 2b is used to decompose the spatial frequency of Moiré fringes into $X_{45°}$ and $Y_{135°}$ and to extract the phase difference between the two directions. The angles of the upper part of the Moiré fringes and the y-axis can be calculated, and the angles of the lower part of the Moiré fringes and the Y-axis can also be calculated. Thus, Moiré fringes and the two gratings are connected with the coordinate axis so that the worktable can be adjusted automatically according to deviation information.

In this method, the phase difference between any two positions in the $X_{45°}OY_{45°}$ coordinate system should be calculated to achieve high-precision alignment, as shown in Figure 5. In the case of the same grating with $P_1$ = 4.0 μm and $P_2$ = 4.4 μm, a theoretical correction accuracy of $10^{-6}$ rad can be achieved [31]. The difference between the upper and lower parts of the fringe can also be detected if an angular offset exists between the imaging system and the fringe. However, after the rotation angle of this method exceeds 2°, the angle generated by the Moiré fringes exhibits a nonlinear relationship with the actual angle, which requires coarse alignment to locate the alignment range.

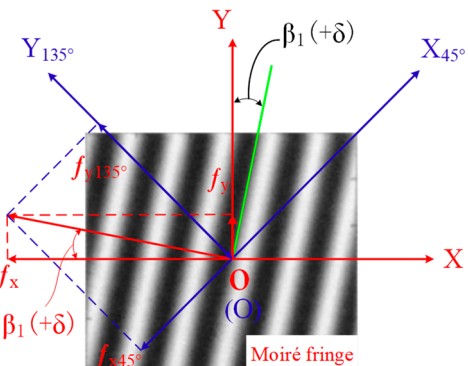

**Figure 5.** Space–frequency decomposition method (SFDM) for correcting angular displacement in plane.

In 2019, Singapore scholar Chan studied and found that a microlens array will form an enlarged Moiré fringe image for overlapping patterns with a rotation angle that can be observed with the naked eye under incoherent illumination [32]. Based on this study, when there are different in-plane angular displacements between the mark on the wafer and mask, the rotation error of the two planes can be determined effectively and quickly by analyzing the Moiré image and its magnification. At present, the detection accuracy of this method is low, making it only suitable for the coarse alignment detection of angular displacement. The differential marking pattern design and the matching algorithm of images with different magnifications are expected to further improve its accuracy.

As stated in Section 3.2.1, at the single point where angular displacement can reach the detection accuracy, the wafer edge will still have a large deviation, so single-point detection has little significance for global alignment. Multi-point angle detection in the plane is required, and global alignment of the rotation axis at different positions of the wafer can be achieved.

### 3.2.3. Fine Displacement Correction after Grating Parallelization

After calibrating the inclination and rotation angles of two planes, line grating imaging is performed to achieve high-precision displacement correction in the vertical grid line direction.

In 2012, Zhu et al. of the Institute of Optoelectronics at the Chinese Academy of Sciences proposed a four-quadrant grating alignment method for near photolithography based on phase demodulation [33], as shown in Figure 6a. According to the fine calibration principle of Moiré fringes in the vertical grating direction, the displacement between the mask and wafer causes a spatial phase change of Moiré fringes, and phase demodulation is performed to achieve accurate axial alignment.

The center of the four-quadrant grating is added with a line width of 20 μm for coarse alignment to position the line grating within the range of fine alignment; an enlarged view of one mark is shown in Figure 6b. Because the line grating can only amplify the displacement perpendicular to the grating direction, four corners are used to arrange intersecting gratings to realize the displacement by the capture of the lower *x* and *y* axes in the two-dimensional plane.

The range of sensitivity can be extended by using gratings with various periods for four-corner alignment markings. The movement directions of the upper and lower Moiré fringes are opposite, and the alignment accuracy can be doubled to achieve measurement errors below 10 nm. After the correction of axial deviation by line grating, fine alignment adjustment detection and pre-sequence secondary compensation correction are required to ensure the reliability of alignment accuracy and to reduce the interference caused by the post-sequence adjustment. Studies have revealed that the integer cutting of the period of the Moiré fringe image can improve the accuracy of the phase extraction algorithm.

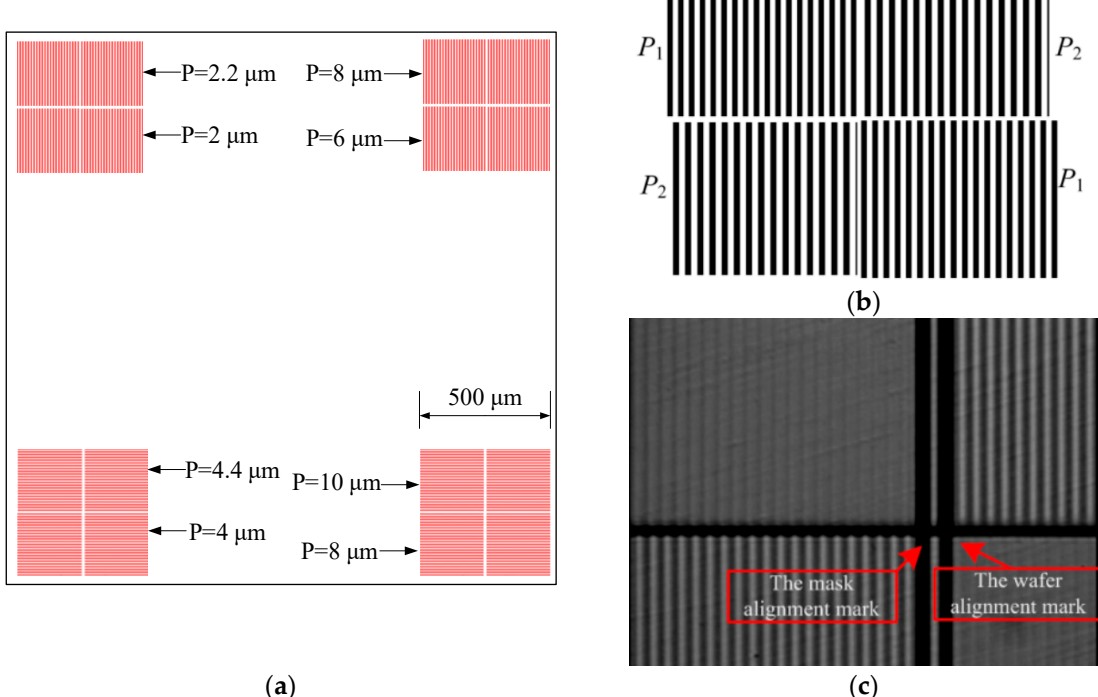

**Figure 6.** Four-quadrant gratings. (**a**) Complete alignment mark. (**b**) Enlarged view of complete mark. (**c**) Moiré fringe image.

The design of alignment markers has been the most typical application of Moiré fringe photolithography alignment since 2010, and it has been subsequently optimized.

## 4. Accuracy Analysis of Moiré Fringe Lithography Alignment Technology

The alignment method determines the range of alignment accuracy of the lithography system. The alignment accuracy of Moiré alignment technology can reach the sub-nanometer level. However, in the actual lithography alignment process, the working environment, operating equipment, processing algorithms, and other factors considerably affect accuracy. Therefore, conducting an in-depth analysis of the factors affecting the alignment accuracy is critical for eliminating controllable factors and offsetting uncertainty factors so that Moiré alignment accuracy in the actual lithography system gradually approaches the upper limit of theoretical accuracy.

According to the introduction and analysis in Section 3, the main factors affecting the alignment accuracy of Moiré fringes are Moiré fringe generation, alignment process, fringe imaging, and post-processing, which are analyzed separately in this section.

### 4.1. Generation Stage of Moiré Fringes

Moiré fringes can be affected by grating, light source, and marking structures.

(1)    Structural parameters and fabrication process of the grating:

The magnification of Moiré fringes for small displacements is inversely proportional to the grating period difference between the two gratings. Considering the difficulty of the grating manufacturing process, manufacturing errors are inevitable. Theoretically, the smaller the period difference, the better the magnification. However, when the magnification is too high, the frequency of the generated Moiré fringes decreases, and the difference frequency signal is closer to the zero-frequency signal, which reduces phase extraction accuracy and the range of the linear relationship of displacement amplification. The edge information of the fringe image is lost, which reduces the signal-to-noise ratio. When the period ratio of the two gratings is close to 1.1, the magnification and Moiré image quality

can be compromised. The total number of lines and the grating depth considerably affect the measurement accuracy.

(2)    Type, wavelength, and accuracy of the alignment light source:

The phase distribution of Moiré fringes is highly sensitive to the divergence angle of the incident light source; thus, the type, wavelength, and collimation of the alignment light source directly affect the alignment accuracy. Compared with a common light source, a parallel light source facilitates easy identification of the alignment state [34]. Dual dipole illumination can reduce the blooming light generated by the light source at the mark edge and is more suitable for the Moiré fringe photolithography alignment system.

(3)    Alignment marking design:

The structure of the alignment mark determines the form and position of the generated Moiré fringes. The targeted design can effectively utilize the grating characteristics and can realize functions such as expanding the application field, optimizing the alignment process, expanding the alignment range, and improving the alignment accuracy.

### 4.2. Alignment of Mask to Wafer Stage

During the actual alignment of the wafer and mask, the periodic repeatability of the Moiré pattern and the different positions of two marks in the mask and wafer can affect accuracy.

(1)    Periodic repeatability of the Moiré fringe:

The periodic repeatability of Moiré fringes limits the range of fine alignment to one cycle. Low-frequency coarse alignment marks are arranged next to fine alignment grating, and the fine alignment Moiré fringes are imaged. Coarse alignment is achieved by analyzing the images of the low-frequency grating. The spectral distribution of the two groups of fringes is independent of each other to achieve coarse and fine synchronization alignment, which can partially mitigate this effect.

(2)    The Talbot effect:

When the grating is at various orders of the Talbot distance, the contrast of Moiré images differs considerably, and at odd Talbot distances, a phase shift occurs, which interferes with the phase extraction of subsequent Moiré fringes [35]. Although the change of the gap does not affect the visibility of the Moiré fringes within a certain range, for the phase extraction of the high-precision Moiré fringe alignment, the gap of the two gratings should be set according to the Talbot distance to obtain images with superior contrast.

(3)    Parallelism between the mask and the wafer

After the position of the wafer is determined, the grating marking on the substrate and the alignment marking on the mask typically exhibit certain angular displacement and out-of-plane inclination. During phase extraction, the alignment marking of the mask and wafer should be parallel; otherwise, the phase extracted would be highly erroneous.

### 4.3. Moiré Image Generation and Post-Processing Stages

Once Moiré fringes are formed, the ways in which their images are received and processed have an impact on accuracy.

(1)    Performance parameters of CCD:

The resolution of CCD and the ratio of the grating line width to the size of the CCD pixels affect the correct extraction of the Moiré fringe phase, and the position of CCD at axial and transverse distances and the degree to which the grating line is parallel to the *x*- or *y*-direction will change the visibility of Moiré fringes.

(2)    Spectrum filtering algorithm:

The Moiré fringe phase characteristics can be fully utilized by filtering the diffracted light of different orders. The spectrum of a Moiré fringe image includes not only the background intensity spectrum, the fundamental frequency spectrum, and the high-order harmonic spectrum but also the differential frequency spectrum and the sum frequency spectrum. Spectrum filtering can filter prominent Moiré fringe spectrum components, reduce the influence of other spectrum components and various noises, and obtain high contrast and signal-to-noise ratio Moiré fringe images. Therefore, the algorithm of spectrum filtering considerably influenced accuracy. For example, in out-of-plane tilt correction, symmetrical (m, 0) interference emitted from similar angles are mostly used, whereas in-plane rotation angle detection mostly uses (m, −m) interference levels, which can effectively avoid the influence of uncertainties such as gap changes during subsequent lateral alignment deviation correction. In-plane precise displacement calibration is typically achieved by using the (+1, −1) order interference to achieve high-accuracy alignment.

(3)    Phase extraction algorithm

Commonly used methods for phase extraction of Moiré fringes are fast Fourier transform (FFT) and wavelet transform (WT). Compared with common algorithms, a modified alignment system algorithm can achieve superior results. Many algorithms, such as non-linear, linear fitting, and optimal expansion, have been introduced and used to improve the accuracy and efficiency of phase extraction. For example, SFDM is used to measure the in-plane rotation angle. The phase average algorithm (PAA) realizes in-plane displacement correction of double gratings. The synthetic wavelength temporal phase retrieval algorithm(STP) realizes the dual-wavelength interference Moiré fringe scheme [36].

(4)    Image processing algorithms

The fabrication defects of grating, noise, and distortion during wafer fabrication, the photoresist on the wafer surface, and absorption of reflected light by the process layer affect the quality of the Moiré fringe image. Therefore, to reduce the influence of the image on the accuracy, image denoising, and enhancement, low-pass filtering should be performed.

### 4.4. Other Factors

In addition to the three main stages of Moiré fringe generation, imaging and post-processing, other influencing factors exist in the lithography system. These factors include working instruments, such as the step error of piezoelectric ceramics, piezoelectric ceramic driver accuracy, and mechanical instability of the working platform; the optical paths structure, such as the imaging distortion of lens composition and the resolution of optical microscopes; and noise interference, such as environmental noise and electronic equipment noise.

## 5. Optimization of the Moiré Fringe Alignment Scheme

The most critical factors affecting alignment accuracy include marking structure, alignment scheme, and phase extraction algorithm. Moiré fringe alignment has been optimized in all aspects to widen the application range and reduce the complexity of the scheme. This section is intended to analyze how to enlarge the alignment range, enhance the image contrast, improve the flexibility of the scheme, and improve the alignment accuracy.

### 5.1. Extending Moiré Fringe Alignment Range

Because of the periodic repeatability of Moiré fringes, the alignment range of Moiré fringes lithography is considerably limited. Rough alignment is required to locate the position deviation within the precise alignment range.

However, several problems can occur. Coarse marking is typically located next to the grating marking, which occupies a larger area and should integrate the grating images of various positions, which in turn increases the workload and difficulty of the phase analysis. In addition, the spectral resolution is determined by the size and number of fringes. Additional motion is required for the two gratings to obtain a sufficient resolution

of the Moiré fringe image. The separated coarse and precise alignment imaging will increase system complexity. To solve the two aforementioned problems, combining the two markers as a whole can effectively expand the alignment range and simplify the system structure.

In 2015, the University of Chinese Academy of Sciences achieved an alignment range of 120 μm using the least common multiple of the Moiré fringes widths [37]. As shown in Figure 6c, the Moiré fringes in the left and right parts of the image have the same period, and this study generated left and right Moiré fringes with different periods by calculating the least common multiple of the difference between the two periods to achieve wide range alignment.

In 2019, Chinese scholar Tang et al. proposed a large-area composite grating. The main purpose of this method was to make full use of the various directions of the coarse and fine alignment marks so that these can be well distinguished in the spectrum to achieve greater alignment [38]. However, this method considerably increases the area and manufacturing difficulty of alignment marks. In the same year, Huang et al. used eight sets of gratings arranged orthogonally. When the offset distance of the two gratings is too large, the overlapping gratings in the central area do not produce Moiré fringes, which form a cross-shaped image for rough alignment and expand the alignment range. When achieving multi-directional alignment [39], the precise alignment accuracy is low. In addition, the alignment range does not exceed 200 μm in both schemes.

In 2020, China Southwest Jiaotong University proposed a dual-frequency Moiré fringe alignment scheme using the alignment marks shown in Figure 7.

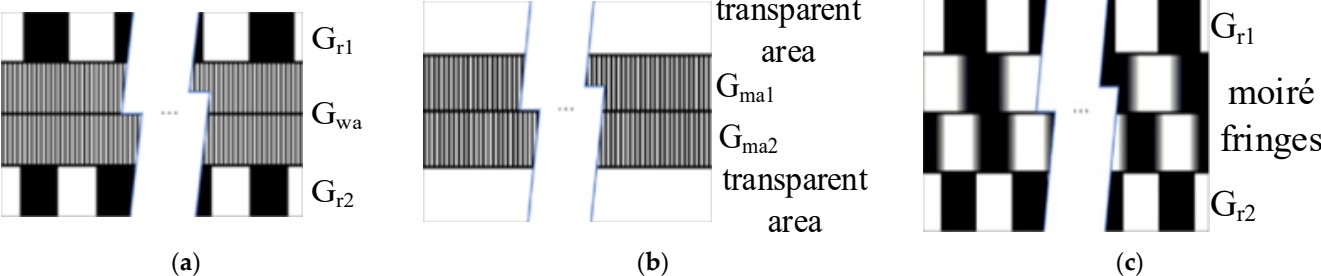

**Figure 7.** Alignment mark and Moiré fringe image. (**a**) Wafer alignment mark. (**b**) Mask alignment mark. (**c**) Moiré fringe image.

In the alignment marks, $P_{ma1}$ = 9.5 μm, $P_{ma2}$ = 9.6 μm, $P_{ma3}$ = 9 μm, $P_{r1}$ = 171 μm, and $P_{r2}$ = 144 μm. Using the Moiré fringes generated by the upper and lower gratings and the positional relationship of $G_{r1}$ and $G_{r2}$ to achieve rough alignment, $G_{ma1}$, $G_{ma2}$, and $G_{wa}$ exhibit periodic differences, which can form differential gratings to form Moiré fringes to achieve fine alignment. The Moiré fringes generated by the upper and lower parts have different periods from $G_{r1}$ and $G_{r2}$ and achieve a wide range of alignment. An alignment range of 500 μm and an alignment accuracy of 10 nm were obtained [40]. When expanding the alignment range, this alignment scheme reduces the steps of setting coarse alignment marks to complete the positioning. Coarse alignment positioning and fine alignment correction are performed in the same image to optimize the alignment process.

### 5.2. Enhancing Moiré Image Contrast

In some lithography technologies, such as ultraviolet nanoimprint lithography and soft lithography, special process layers should be added to the surface of the mask and wafer, which exhibit similar optical properties to the mask material, which change the grating transmittance and reduce the diffraction efficiency and image contrast, resulting in lower phase extraction accuracy.

To improve the contrast of the Moiré fringe image in special process technology, in 2015, Chinese scholars Qin et al. proposed coating an opaque area of the grating on the impression mask with dense optical material [41]. Comparative experiments with different materials of different thicknesses showed that the mask coated with 20 nm of metallic chromium

had the best effect. Without coating, the contrast was only 17%, whereas, after coating, the contrast was increased to over 95% to achieve high accuracy alignment. However, an actual alignment application was not considered for Moiré stripe lithography alignment.

Beginning in 2017, Kikuchi of Japan's Tohoku University replaced the UV-curable liquid between the mask and wafer gap in nanoimprint lithography with a fluorescent liquid layer consisting of fluorescent glycerin and Rhodamine 6G-C1. Under a 585-nm light, the fluorescent markers became excited. The formation of additive Moiré fringes can be observed by the grating excited by the fluorescent liquid. The phase demodulation of fluorescent Moiré fringes is consistent with common fringes. The alignment scheme and the resulting Moiré fringes are shown in Figure 8.

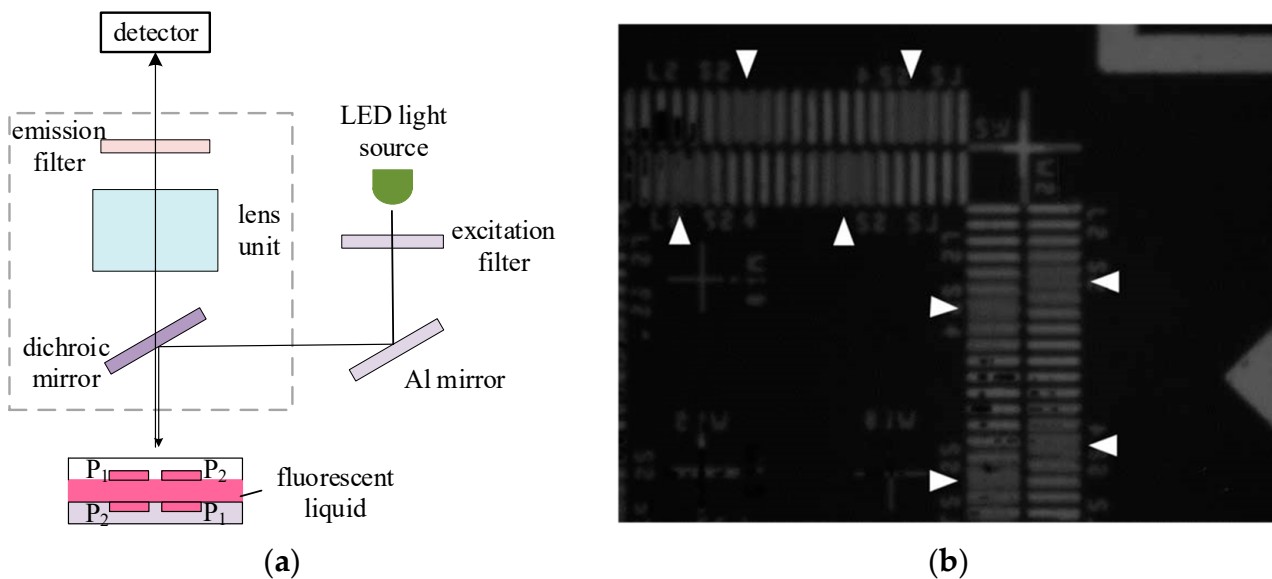

**Figure 8.** Fluorescent Moiré fringe alignment. (**a**) Alignment system schematics. (**b**) Moiré fringe image.

In this method, the excitation of the fluorescent liquid functions as a light source to generate gratings superimposed to form additive fluorescent Moiré fringes. When periods $P_1$ and $P_2$ are 4.0 and 4.4 µm, respectively, and the width of the filled fluorescent liquid is 2.0 µm, the fluorescent Moiré images with excellent contrast and visibility. Corrections of angular displacements from 0.44° and 72.09° to 0° were achieved. In in-plane axial displacement correction, the team demodulated the resulting high-frequency signal and achieved an axial correction with a deviation of 5.9 nm [42]. The uniformity of the residual layer formed by the liquid sandwiched between the mold and the substrate surface can be observed, and its thickness can be monitored.

### 5.3. Improving the Flexibility of Moiré Fringe Alignment

The application of digital grating can reduce the manufacturing cost of physical grating in Moiré fringe alignment, enhance the flexibility of alignment marking design, eliminate the requirement of a complex grating interference system, and provide a convenient and flexible method of synthesis. The application of digital Moiré fringe alignment can be classified into two categories. The first category is the Moiré fringe formed by the grating on the digital mask projected by DMD and the physical grating on the wafer. The second category is the digital Moiré fringe generated by the grating image on the mask and wafer through the computer. Shao et al. proposed to use the grating image and digital Moiré fringe technology for fine alignment of Moiré fringes. The schematic is shown in Figure 9.

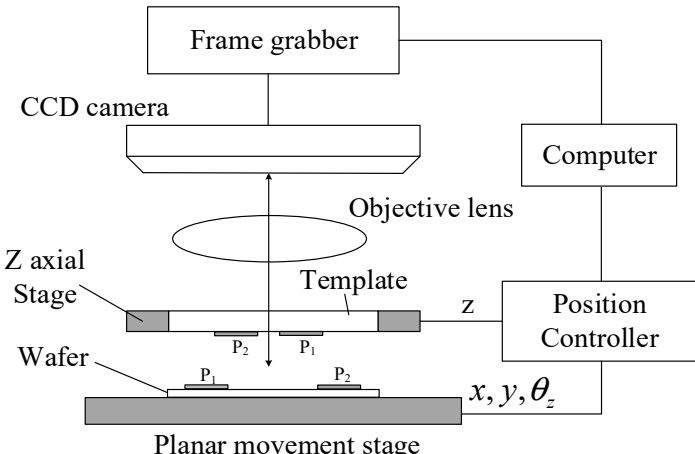

**Figure 9.** Schematic of the digital Moiré fringe alignment principle.

This alignment scheme does not include the laser alignment light source. Only the grating image recorded by CCD generates multiplicative Moiré fringes through computer simulation. The phase analysis method of digital Moiré fringes is consistent with the Moiré fringes generated by the superposition of the aforementioned physical gratings. The two-dimensional FFT and the phase unwrapping algorithm were used to measure the angular displacement. When the two planes are parallel, the relative displacement is measured by extracting and analyzing the phase difference in the frequency domain, and in-plane displacement correction is realized. The alignment accuracy is better than 10 nm [43]. Computers were used to capture grating images to generate Moiré fringes, which can flexibly adjust the mode of superposition and filtering; Moiré fringes generated under various conditions can easily improve accuracy. Moreover, the Moiré fringes generated by computers are less affected by the environment, and phase extraction is easy.

In one study [25], digital grating and physical grating were superimposed to achieve alignment, but the Moiré fringes were still produced by superimposed natural light fields. However, the design of the digital mask was adopted, which reduces the cost of making mask patterns and mask alignment marks.

*5.4. Improving the Alignment Accuracy of Moiré Fringe*

Moiré fringes produced by the superposition of line gratings exhibit high accuracy. However, because of the limitations of various factors discussed in Section 4, breaking through the nanometer level is difficult. Only a few alignment schemes can achieve an alignment accuracy of 3 nm. Most Moiré fringe lithography technologies cannot satisfy the manufacturing conditions of the most advanced semiconductor chips. Improving the alignment accuracy of Moiré fringes to the sub-nanometer level has become challenging. In the case of the inherent obstacles of Moiré fringe alignment, the alignment accuracy is improved to the sub-nanometer level by lithography alignment based on self-coherent Moiré grating fringes and Moiré fringe alignment based on deep learning.

5.4.1. Lithographic Alignment Based on Self-Coherent Moiré Grating Fringes

Lithographic alignment based on self-coherent Moiré grating fringes is a lithography technology combining diffraction light intensity information and Moiré fringe phase information. This method was proposed by the Shanghai Institute of Optics and precision machinery at the Chinese Academy of Sciences in 2017 [44]. The alignment principle is illustrated in Figure 10.

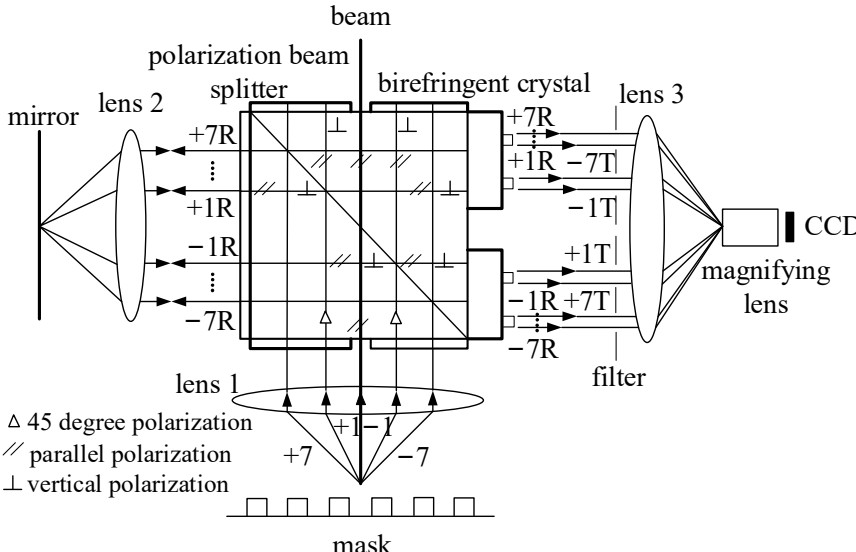

**Figure 10.** Schematic of self-coherent Moiré grating fringe alignment technology.

In this method, the diffracted beams of the same order of the phase grating are divided and transformed to form interference fringes with different periods and subsequently superimposed to generate Moiré fringes, which are aligned by analyzing Moiré fringes. The interference fringes are extremely sensitive to the position change of the grating, and the alignment accuracy is improved by the amplification effect of forming Moiré fringes, which can reach an alignment accuracy of 0.07 nm and an alignment repetition accuracy of 0.11 nm. Although this alignment method improves the alignment accuracy to the sub-nanometer level, the complexity of the system is increased, and the position error of the mask and wafer cannot be determined simultaneously, causing it to lose the advantages of the conventional Moiré fringe alignment, such as the insensitivity to the mask wafer gap and the correlation of two planes' synchronous alignment.

### 5.4.2. Moiré Fringe Alignment Based on Deep Learning

To eliminate the nonlinear and non-additive effects of inherent errors and random errors in practical alignment, Chinese scholar Wang et al. proposed a Moiré fringe lithography alignment method based on deep learning in 2021, and an alignment accuracy at the sub-nanometer level can be achieved by using micron-level grating.

The method involves using a deep learning bias regression measurement framework similar to the VGG network to fully learn the mathematical mapping relationship between the Moiré position information and the actual offset. The network framework is shown in Figure 11. The data set is to use gratings with periods of 4.0, 4.4, 8, and 10 μm, a step size of 0.1 nm, and a 2-μm displacement range to generate 20,000 Moiré fringe images containing displacement phase information for network training. The displacement prediction of the test set was compared with the experimental results of the classical FFT and WT (Table 1).

The experimental results revealed that the deep-learning-based bias regression measurement strategy considerably outperforms the FFT and WT processing methods and achieves sub-nanometer alignment accuracy [45]. Subsequently, the zero-elimination squares were added to the test set image to simulate the manufacturing defects of the grating, and different numbers of fringes were cropped. Various noises were introduced for comparative analysis. Experimental results revealed that the errors of this network model were considerably lower than those of the classical algorithm, and the model exhibits excellent robustness to manufacturing defects, environmental noise, frequency leakage, and systematic errors.

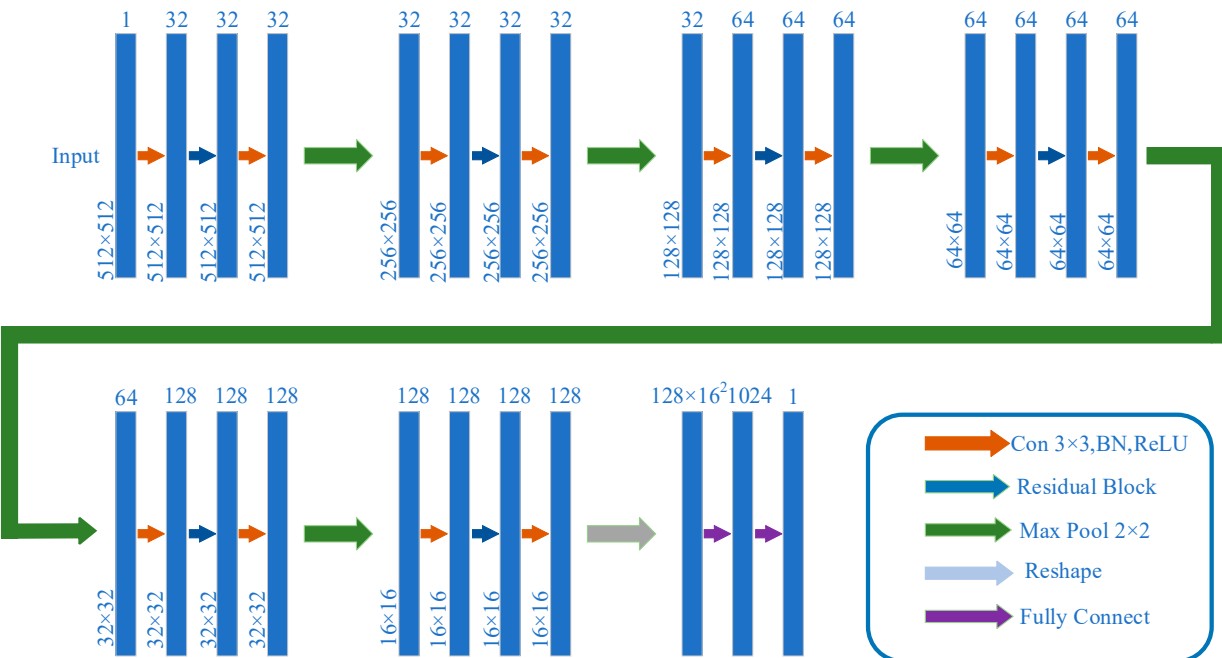

**Figure 11.** Deep learning-based bias regression measurement network framework.

**Table 1.** Error comparison between deep-learning-based bias regression measurement strategy and classical algorithms.

| Method | P1 = 4 μm and P2 = 4.4 μm | | | P1 = 8 μm and P2 = 10 μm | | |
|:---:|:---:|:---:|:---:|:---:|:---:|:---:|
| | MaAE | MeAE | SD | MaAE | MeAE | SD |
| FFT | 13.05 | 5.67 | 7.96 | 13.13 | 5.85 | 7.03 |
| WT | 12.65 | 5.58 | 6.86 | 12.76 | 5.76 | 7.85 |
| DLMRM | 0.60 | 0.22 | 0.42 | 0.58 | 0.23 | 0.31 |

Note: MaAE, MeAE, and SD in Table 1 represent maximum absolute error, mean absolute error, and standard deviation. The unit of all of these parameters is nm.

However, because the training set of this method only contains two sets of grating parameters with different periods and one grating arrangement structure, the model transfer ability is poor, and its application range is considerably limited.

## 6. Development Trends and Prospects

The results of the aforementioned analysis revealed that the main criteria for measuring the quality of Moiré fringe lithography alignment technology are alignment accuracy, range, and process. However, in practice, the aforementioned standards are typically not satisfied at the same time, and sometimes, conflicts even arise; for example, the high alignment accuracy and the simple alignment process cannot be perfectly satisfied at the same time. Therefore, the development of Moiré lithography alignment technology is not unidirectional. To satisfy various requirements, such as alignment accuracy, manufacturing process, and application scenarios, developing targeted Moiré lithography alignment technology is critical.

The optimization of the three aspects, namely marking structure, alignment scheme, and phase algorithm, can effectively improve the performance of Moiré fringe lithography alignment. The most promising development directions regarding these three aspects are as follows:

(1) Marking structure:

Optimizing the marking structure can simplify the alignment process and reduce the steps of in-plane, out-of-plane, and axial alignment. Therefore, the three-dimensional

alignment can be completed in one alignment operation, eliminating the interference in the step-by-step alignment. Furthermore, if smaller period gratings are introduced into conventional alignment marks, this may achieve sub-nanometer alignment accuracy.

(2)     Alignment scheme:

Using digital grating as an alignment mark and combining with Moiré fringe digital synthesis technology, a fully digital lithography alignment scheme is proposed, which can reduce the mask cost and improve design flexibility.

(3)     Phase algorithm:

On the basis of the existing deep learning model, the database samples can be expanded; model transfer training and deployment can be performed to realize automatic alignment, which can improve alignment efficiency while improving alignment accuracy.

Lithography technologies, such as nanoimprint lithography, surface plasmon lithography, and dielectric microsphere imaging lithography, can overcome the limitations of the diffraction limit and can become the next-generation mainstream lithography technology. Moiré fringe alignment technology has good compatibility with the abovementioned lithography technologies and will certainly have better development prospects in the future.

**Author Contributions:** Conceptualization, W.J. and H.W., methodology, H.W.; validation, W.J. and H.W., data curation, W.J. and H.W., investigation, H.W., W.X. and Z.Q., writing—original draft preparation, H.W., W.X. and Z.Q., writing—review and editing, W.J., supervision, W.J., funding acquisition, W.J. All authors have read and agreed to the published version of the manuscript.

**Funding:** This work was supported by the Natural Science Foundation of China under grant no. 61875166, Sichuan Science and Technology Program under grant no. 2021JDJQ0027.

**Data Availability Statement:** The data underlying the results presented in this paper are not publicly available at this time but may be obtained from the authors upon reasonable request.

**Acknowledgments:** W.J. also would like to acknowledge the Sichuan Provincial Academic and Technical Leader Training Plan, the Xihua Scholars Training Plan of Xihua University, and the Overseas Training Plan of Xihua University (09/2014-09/2015, University of Michigan, Ann Arbor, US). We also thank LetPub (www.letpub.com accessed on 24 February 2023) for its linguistic assistance and scientific consultation during the preparation of this manuscript.

**Conflicts of Interest:** The authors declare that they have no conflict of interest.

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
