# Peer review of "Lithography Alignment Techniques Based on Moiré Fringe"

_photonics, doi:10.3390/photonics10040351_

Round 1

Reviewer 1 Report

In this review article, Jiang et. al. discussed various lithography alignment techniques based on moiré fringes that have important applications in semiconductor manufacturing. This is a timely review that helps readers to understand the alignment principles and processes, the advantages and limitations of each technique, and the challenges involved in ultimately achieving sub-nanometer accuracy. I recommend publication in Photonics after the authors address the following points and revise the manuscript accordingly.

·       Page 2, line 59: The statement "In the eighteenth century, French scientist Moiré discovered Moiré fringes" is false. Moiré does not refer to a presumed French physicist who studied moiré patterns (see ISBN: 978-1-84882-180-4, Chapter 1). Please be careful and check for factual errors in the manuscript.

·       Some sentences are unnecessarily long, e.g. “In 1991, Asundi et al. of the University of Hong Kong… In 1998, Roberto of the National University of La Plata… In 2012, Tang et al. of the Institute of Optoelectronic Technology of the Chinese Academy of Sciences…” (Page 4, Lines 161 – 167). I think that the author institutions of the cited works can be omitted as they are not directly relevant to the discussion.

·       Page 20, line 732 – 733: “Furthermore, if sub-wavelength gratings can be introduced into conventional alignment marks, this may achieve sub-nanometer alignment accuracy”. How to observe moiré fringes or verify sub-nanometer alignment accuracy if the grating period is smaller than the diffraction limit of light? Please discuss.

·       Does the grating duty cycle affect the alignment accuracy and/or the contrast of the moiré fringe? Please discuss.

·  About using moiré images for coarse alignment, doi.org/10.1021/acsnano.9b06772 might be an interesting reference.

Reviewer 2 Report

This paper is a collection of discussions about works previously published by various other authors. There is no new material. The discussions are hard to understand, contain inaccurate or useless information, and are sometimes self-contradicting.

Although unclear, some of the discussions appear to measure the rotational angle and the tilt between the mask and the wafer. Although this is in principle important for site by site alignment, it is unnecessary with the precision stages available for global and modified global alignment. Moreover, aligning to a few nm at both sides of a single chip a few mm wide results in a rotational accuracy on the order of 10-6 radians, 100 times better than claimed in this paper. Viewed another way, such a chip perfectly aligned at one side with the paper’s stated rotational error would be misaligned at the other side by hundreds of nm. The tilt “correction accuracy is better 10-3 radians” which implies a height difference of 300 microns from one side of a 30 cm wafer to the other.  Rotational and tilt alignments are pointless for modern integrated circuit lithography.

Other discussions refer to digital Moire fringe alignment. The discussions are unclear, and the figures show only the chip (Fig. 2a), only the mask (Fig. 11), or both a template and the wafer (Fig. 10). Viewing the chip only could be useful in global or modified global alignment as a supplement to the precision stages, but the position of the mask with respect to other chips on the wafer must also be known.

Detailed technical comments by line or figure number

51. Reference to magnetic susceptibility is unclear.

78. Paragraph is unclear.

113. “vertical” direction is unclear.

121. The following paragraph is unclear without figures.

Figures 1b and 1c. Not adequately explained.

147-151. Alignment marks are no more difficult than other mask patterns, imprint masks do not have short lifetimes, and masks are generally written with e-beams, not lasers or ion beams.

Fig 2a. A dichroic beam splitter either transmits or reflects a given wavelength, not both as shown. It should be drawn at 45o to reflect at 90o.

165. “Taber effect’’ Did the authors mean Talbot effect? Entire paragraph is unclear.

188: I question whether Moire fringes are the “most common method” of alignment.

196. How can ”numerous” angles exist?

200. This section is very hard to follow. No examples are given. As commented above this is unnecessary with modern stages and global or modified global alignment. In addition, I question diffraction analysis since gratings are virtually adjacent and only the zero order is observed.

229. This section is very hard to follow. Same comments apply as for line 200 above. In addition, the effect of a small angle in-plane rotation is a change of the Moire fringe position as a function of the position along the average direction of the gratings.

255. This section is very hard to follow. Same comments apply as for line 200 above.

269. What are “mask wafers”?

285. What does “shortening the light source” mean?

Figure 3 is incomprehensible. A gap is shown underneath the wafer, three unidentified structures are shown intersecting the wafer, and three more apparently form an image.

308. 10-3 radians is an enormous tilt for a wafer, 300 microns across the diameter of a 30 cm wafer. Totally impossible.

Figure 4. Unclear what it shows. Are top and bottom images from tilts in different directions? Why degrees when previously said radians?

323. Now 10-4 radians can be measured but 10-3 radians was too small to incline fringes?

Figure 5. Totally unclear with claimed accuracy of only 10-3 radians.

359-382. Does “mask wafer” mean mask to wafer? Angular displacement should be “calibrated,” not corrected? Irrelevant in global or modified global alignment. 10-4 radians is 30 microns across a wafer, which is enormous.

Figure 7. Unclear. What is where? The text does not help explaining the figure.

464. Maybe “number of gratings” means number of lines. But what does “balanced” mean?

470. What is a “double-dipole” light source, and what happens at the “marked edge” of what?

489. “partially compensate” is unclear

490. Using the Talbot effect would require extremely parallel illumination, which is a severe limitation.

503. The following paragraph makes no sense.

512. What is the “ratio of CCD pixels?

516. The following two paragraphs are unclear.

544. Reference to “aforementioned three main stages” is unclear.

566. Fine and coarse alignment need not “interfere with each  other.”

570. Paragraph is vague.

573. Paragraph is hard to understand.

Figure 8. Totally unclear. Filled with unexplained features.

608. Coating the mask with chromium helped? Why didn’t it make everything, including the features on the mask, opaque?

613. Following paragraph and Figure 9 describe a sensible method of improving the Moire signal, but the insertion of a liquid dye between the mask and the wafer could well interfere with the wafer processing.

Figure 10.  Shows Moire gratings on both mask and wafer that completely miss each other and no path for exposure light to enter and no origin for the Moire light.

Figure 11. Completely incomprehensible, and missing the wafer.

732. “sub-wavelength gratings” may be completely opaque.

Additional non-technical comments.

The authors referred to are identified by nationality, either individually or by the extended title to where they did the work.

No mention is made of the 10 micron proximity X-Ray lithography alignment system developed by H. I. Smith at MIT, I believe in the 1990’s. 

Reviewer 3 Report

The manuscript titled "Lithography alignment techniques based on Moiré fringe" by Jiang et al. reviews the history of the Moire fringe technique for alignment appplications up to current developments involving machine learning. The review gives an comprehensive overview of the method, its development and application. It is generally well written, only minor editing (there were missing spaces from time to time) is suggested before publication.
